# Physical Literacy Assessment Tools: A Systematic Literature Review for Why, What, Who, and How

**DOI:** 10.3390/ijerph18157954

**Published:** 2021-07-28

**Authors:** Habyarimana Jean de Dieu, Ke Zhou

**Affiliations:** Physical Education School of Henan University, Kaifeng 475000, China; jdhdieu@yahoo.fr

**Keywords:** physical literacy, assessment tools/instruments, evaluation tools/instruments

## Abstract

Physical literacy (PL) has been acknowledged to be an individual journey, in view of this contention, objective assessment of such a developing construct has become a debatable issue for the last couple of decades apart from physical domain of observable domain. The purpose of this systematic review was to scrutinise what is currently known regarding the PL assessment tools—the existing PL assessment tools, their pioneers and year of publication, the philosophy behind their initiation, what they have been assessing (assessment domains), the category of population being assessed in relation to their age group, validity of the tools, other scholars notes, as well as the approaches being used, whether assessment for, as or of learning during physical activity participation. The Preferred Reporting Items for Systematic Reviews and Meta-Analysis (PRISMA) was used to undertake a comprehensive search from six electronic databases—ScienceDirect, Scopus, Eric, PubMed, Google Scholar, and SportDiscus retrieved 52 research articles and review papers, whereby only 22 articles were included after identification, screening, and eligibility selection criteria process. The study established that the majority—70%—of PL assessment tools were developed to promote either fundamental movement skills, athlete development or long-term health and well-being, instead of lifelong participation in physical activity. It was also ascertained that only 30% of PL assessment tools address both three domains comprising PL. Of a particular concern, it was explored that only the International Physical Literacy Association (IPLA) PL matrix takes into account everyone of any age group, while the rest of the others segregate participants falling in a specific age groups to be based on. Afterward, the majority of PL assessment tools were identified at 70% to adopt assessment for learning or at a certain time combination with assessment as learning while assessing individuals’ PL progress. The conclusion was therefore drawn that the overall purpose of PL- to value and take responsibility of engaging in physical activity for life is still largely absent among the majority of existing PL assessment tools and both the ideal of what to assess and who to be assessed are far less to be met, while the effective PL assessment approaches remain critical. After all, in light of this conclusion future agenda has been suggested in view of designing PL assessment tools effective enough to promote PL for all.

## 1. Introduction

Despite the fact that either formative or summative assessment reflects an important aspect of pedagogy, it could also be argued that assessment is undertaken for evaluative and accountability purposes instead of celebrating what has been achieved, what is valued by participants, and the way in which progress has been made from a certain point [1]. Thus, it is crucial to understand what PL is, its components, and the effective mechanism of its assessment resulting in its development and promotion [2,3].

The overall purpose of physical literacy is to value and take responsibility for lifelong participation in physical activity [4,5,6]. Not long afterwards, while clarifying the concept of physical literacy, Whitehead [7] a proponent of the PL concept, insisted that PL is not a state to be achieved and maintained thereafter. Rather, it is a journey to be chartered overtime due to the fact that an individual’s PL is considered to be quite unique to the extent prohibiting any comparison of one individual from another’ previous and current PL level. This view is consistent with the previous argument of [8] (p. 4), which expressed that PL is not a skill. Rather, a disposition to use experience, understanding, and abilities to effectively interact with the world. Reflecting on this contention, the act of assessing individuals’ PL is likely a process, not an end result. Thus, a fact indicating that a tool/instrument designed to collect such data requires us to provide an opportunity to celebrate an individuals’ PL progress. 

For any assessment to serve an appropriate purpose in improving learning experiences, ref. [9] clarified that it should match with assessment for learning approach in which assessment is undertaken more than once by creating a description foregrounding every individual’s personal strengths and weaknesses to be used in serving the following stage of learning, and in turn provides the participants with necessary feedback (refers to a statement indicating the recognition of the desired goal, evidence of present performance, and some understanding of the available way to close the gap between the two provided to the concerned participants [10] that will further their learning through checklists, performance rubrics, artefacts, portfolios, worksheets, etc., to trace their progress along the learning continuum. 

For the sake of clearing any ambiguity, particularly for the researchers and scholars [11,12,13], claimed irrelevance of summative assessment against a multifaceted concept of PL. Ref. [14] explained that charting PL progress is more appropriate than PL assessment due to the fact that PL is a personal journey that needs to be charted through individuals’ previous knowledge, understanding, and experiences, instead of comparison with others.

A supportive view for this assertion was articulated by [15], which elucidates that assessment for learning approach greatly serves in seeking and interpreting evidence for use by both teachers and learners for deciding where the learners reach in their learning, where they need to go, and the best way to get there. 

The additional concern could be the philosophical (refers to logic behind something containing the reasons and cases justifying the value of the concept at hand [7] monism, existentialism, phenomenology and conceptual-motivation, confidence, physical competence, knowledge and understanding principals advanced by [4,16] as bedrocks in building PL tools/instruments necessary to trace and improve the progress of PL among the participants—be they children, adults or old people.

In other respects, PL provides the participants with not only an opportunity of lifelong engagement in physical activity, but also positive health benefits {[17] (pp. 83–99) [18,19,20,21,22,23]}. A support for this assertion was articulated by [24], who ascertain health benefits like alleviation of risks of cardiovascular diseases, diabetes, cancers, etc., as an effect of actively participating in physical activity.

Unfortunately, the effective approaches to PL assessment and appropriate tools equipped with both philosophical and conceptual principals in charting progress in PL, aiming at increasing the individuals’ ability of valuing and taking full responsibility to get involved in physical activity for life, remain critical. 

As a matter of fact, ref. [25] warn that a balance exists between the creation and the use of reliable and valid measurements of progress in relation to participants’ PL journey and the development of process, which assesses both the philosophically complex and holistic nature of concept, remain superficial. For example, ref. [26] report that the availability of relevant PL assessment tools to collect data about children’s development remains an unanswered issue. Besides, both [27,28] comment that, although most of the existing PL assessment tools attempted to assess PL progress, the adopted simplistic, linear, and reductionist tools lack the essence of PL. Accordingly, ref. [29] note that PL assessment over the last two decades has been undertaken to measure only the level of fundamental movement skills or fundamental sports skills.

A supportive view was also articulated by [30,31] who asserted that PL has been assessed focusing much on fundamental movement skills and skill competency. In addition, ref. [3] argue that PL has been assessed through the approaches of either assessment of or for learning, since it has been understood in relation to health/physical education. In a similar vein, ref. [2] identify a scarce of PL assessment tools developed the capacity to collect progressive evidences of an individuals’ PL under the distinct and interlinking domains-affective, cognitive and physical, in case lifelong participation in physical activity is the solitary aim of PL. To the same extent, ref. [32] advocates that a shift from assessing success against norm referenced standards to assessing progress against criterion referenced and embracing the holistic nature of the concept, be taken into account while assessing/charting individuals’ PL journey. 

Such a variety of methods and components in PL assessment have been explained by several researchers that a certain PL assessment tool measures the components actually described in the PL definition and philosophy believed to be effective by the founder organisations or scholars in relation to their policies and culture [2,33]. This modification of PL definition has been reflected on by [7] who comments that realignment of PL definition does not matter, in case it maintains its monism philosophy, whereby both cognitive, affective and physical components are fully and equally maintained. Along the same lines, ref. [25] argue that assessing/charting PL depends on the way in which it is defined and operationalised. Running parallel with this point, ref. [34] elucidated that policies and practices developed to promote PL, particularly in teaching and learning approaches and monitoring and assessment of PL, all depend on the way PL is defined. Thus, ref. [33] comment that much remains to be done in realigning PL definition, philosophy and measurement. Different PL definitions used to underpin the existing PL assessment tools have been explored. 

Consequently, the prevalence of physical inactivity among all categories/age groups of population across the world remains high. Accordingly, ref. [35] declared sedentary health style (physical inactivity) a fourth leading factor for mortality. A fact indicating even that the prevalence of hypokinetic diseases and other sedentary health style-based problems like mental health problems and disorders remain a pressing concern. 

To take a case in point, the study conducted in 2019 by [36] points out that the majority—81% of children and adolescents are not physically active, of which 77.6% and 84.7% are boys and girls, respectively. In a similar light, ref. [37] reports that only 20% of the world’ adolescents were adequately physically active. More exactly, over 90% of school aged children and adolescents in China are classified insufficiently physically active [38]. Of a particular concern, over 340 million of children and adolescents were classified as overweight or obese in 2016 [39], 470.0 million of adolescents were diagnosed with diabetes mellitus in 2017 [40], and it has also been reported that 16% of the global burden of disease and injury in adolescents aged 10–19 years old is due to mental health conditions [41]. Of a related concern, is the high cost—health care costs, lost productivity, and premature mortality of physical inactivity to the countries’ economy like annual estimation of $13 billion in Australia, $117 in USA [42,43]. 

This evidence is consistent with the rationale pushed by Prof. Margret Whitehead to stand up and conceive the concept of physical literacy, namely: the interaction of human being and the environment through movement, the need for fundamental movement skills in early years, hypokinetic diseases plague the nowadays children and adolescents, and the recent custom of providing physical activity only for elites [4]. Consistent with this view is an assertion articulated by [44], arousing the community that to create an active and health population in light of preventing generation of children to grow up with chronic health problems requires establishing a PL foundation. 

To this end, this current review scrutinised what is currently known regarding the existing PL assessment tools, their pioneers and year of publication, the philosophy behind their development, what they are assessing (assessment domains), the categories of population being assessed in relation to their age-group, validity of the tools, other scholars notes, as well as the approaches being used, whether assessment of, for or as learning during physical activity participation. 

## 2. Materials and Methods

This systematic review was carried out by following the reporting checklist of the Preferred Reporting Items for Systematic Reviews and Meta-analysis (PRISMA) guidelines for systematic reviews [45]. For the purpose of this study, a comprehensive search was undertaken to identify related papers. All details of this systematic review were successfully submitted for registration in the International Prospective Register of Systematic Reviews (PROSPERO) under the registration number 269,400 for ensuring that this systematic review is within scope and that the fields have been completed appropriately. 

For the purpose of documenting the analysis method and inclusion criteria, a protocol was developed in advance. Thus, a search strategy was developed in order to identify relevant literature containing the key terms in their title, abstract, and/or key terms. This search strategy was tailored to six electronic databases: ScienceDirect, Scopus, Eric, PubMed, Google scholar and SportDiscus. The search terms used with Boolean Operators were: “physical literacy” AND “assessment tools OR instruments” OR “evaluation tools OR instruments”. The last search was performed on 12 March 2021. 

The selection criteria was based on the PRISMA statement [45]. The search mainly focused on the mapping existing literature on physical literacy assessment tools in the field of social sciences, health sciences. The search then was narrowed to the subject areas, namely: sports pedagogy; social science; sports, recess, recreation and dance fields. The search span was subject to the papers published from the years 2000 to 2020. All articles published before 2000 were excluded from the search. Figure 1 shows the literature inclusion and exclusion at every stage (PRISMA statement). 

The study is only based on original research articles and review papers. For maintaining the quality of review, all duplications were thoroughly checked out. Abstracts of the articles were deeply checked for the analysis and purification of the articles to ensure the quality and relevance of academic literature included in the review process. An evaluation of each paper was carefully undertaken at the next stage. Subsequently, for the limitation of the papers published only in English, the next exclusion criteria was carried out. Then, 2 non-English article publications were excluded from the study. Moreover, after the filtration of duplicate records, 10 more articles were removed from the study due to the fact that they did not describe the assessment mechanisms of the tool(s), but addressing the tool(s) adopted in the study to assess PL in their interventions. The extraction of 22 articles was recorded after assessing each article from the aforementioned inclusion and exclusion criteria. 

During the data extraction phase, 22 articles were selected and the extraction characteristics were: first, originality of the research article, review paper. Published reports, concept notes, conference papers, case studies were excluded. Second, the articles written in English and undertaken from the field of social sciences and health sciences. Third, the articles published between 2000 and 2020. Fourth, the articles aimed at exploring the standardization or development or assessment approaches of the existing PL assessment tools.

## 3. Results

### 3.1. Research Articles and Review Papers 

The current study reviewed 22 research articles and review papers. The study selection process has been summarised in Figure 1. The literature search through the databases resulted in 52 records, and three articles were eliminated due to the fact that they had been published before 2000. The full texts of the remaining 49 articles were carefully screened, and 27 articles were excluded, since they did not meet the eligibility criteria.

### 3.2. Physical Literacy Assessment Tools 

The results of Table 1 shows a 10 PL assessment tools conceived by different public or private organisations, scholars and researchers interested in physical activity, physical education and physical literacy and benefits claimed to be associated with them like quality of life, socialisation, executive function (the functions that help to execute thinking and cognition, including planning and organisation, shifting tasks and knowledge, manipulating information held in working memory, inhibiting inappropriate responses and using context to evaluate the appropriateness of responses [46] just to mention a few). At the same time, it indicates the corresponding pioneers of such PL assessment tools and year of their publication, their assessment domains, target audience to be assessed in the form of age groups, the overall reason behind their initiation, validity and the corresponding comments of the different researchers. For further understanding of different assessment domains and philosophies adopted to guide assessment while using these tools, Table 2 clarifies the PL definition used to underpin these PL assessment tools at hand.

Table 2 shows all 10 different definitions adopted by the pioneers of the existing PL assessment tools which in turn guided the assessment domains of each tool. The issue to note is with regard to the definition advanced by the current authors in view of reflecting both philosophical and conceptual PL underpinnings. As such, PL refers to a monism, existentialism, and phenomenology philosophical based disposition conceived to curb global physical inactivity while increasing lifelong participation in physical activity by promoting the motivation and confidence, knowledge and understanding, physical competence of all people irrespective of their age group or living place. 

## 4. Discussion

The results of Table 1 shows 10 existing PL assessment tools already developed and in use of PL assessment to determine PL of different categories of people in respect to their age groups in light of promoting the progress of physical literacy across the world. 

### 4.1. What Is Being Assessed 

Initially, the current authors were interested in identifying what the existing PL assessment tools are measuring (assessment domains). From the research results, it was indicated that assessment domains are different from one tool to another whereby most of them mainly assess one domain or combination of two, while few of the remaining assess the holistic nature of three domains of PL. In this regard, FMS: 60 Minutes Kids Club (60 MKC) PL assessment, CMOPL and PL Observatory tool (PLOT) have been identified to mainly assess physical competence. On the other hand; PLAY, National standard for K-12 physical education and PE metrics (SHAPE) and CAEPL were revealed to assess the combination of physical competence and cognitive domains; PPLI to assess the combination of affective and cognitive domains. While the minority 3/10 equivalent to 30% of PL assessment tools—IPLA, PFL and CAPL assess the whole three domains of PL. 

Of a particular concern, it is worth mentioning at this point that though CAPL was classified as one of the tools that measures three domains of PL as a whole, HALO’s dualistic philosophy of treating the body as an object that is meant to achieve an acceptable level of performance on a predetermined scale, whereby CAPL put much emphasis on assessing physical fitness part. Hence, there is a contradiction in monism philosophy, which is the bedrock upon PL concept [31]. Thus, a critical issue for CAPL is to fit in Whitehead’s PL, which narrowly addressed physical fitness [4,14]. 

These findings showed that the majority of existing PL assessment tools have been assessing attributes of physical literacy under either one or two domains while marginalising the rest of the others, rather than addressing multifaceted interdependent domains equally. These data are inconsistent with several research findings that established that a significant relationship exists between PL components—*motivation, confidence, physical competence, knowledge and understanding*—on physical activity participation [71,72,73,74,75,76]. A fact which indicates a long way ahead for PL promotion. 

This philosophy was therefore against that of [4], who emphasizes that physical literacy progress should be charted under monism philosophy, which postulates that a human being is naturally made up of three interdependent, interrelated or intertwined cognitive, affective and physical domains. Subsequently, such a finding contradicts the assertion of [31] who insist that an individual’s PL progress assessment be undertaken through a combination/integration of motivation, confidence, knowledge and understanding in relation to his/her embodied interaction with the environment. 

More emphatically, this finding was against the idealistic perspective/academic approach established by [77] who discussed such idealistic perspective, which argues that PL is a holistic concept, i.e., the three domains—affective, physical and cognitive should remain inseparable. Hence, assessing PL in a given way that these domains are separately addressed or ignoring some of them, would contradict the holistic philosophical underpinnings of PL concept. 

### 4.2. The Reason Why These PL Assessment Tools Have Been Developed 

In this second place, the authors sought to ascertain the philosophy adopted in developing these PL assessment tools. In this light, some of them—(60 MKC) PL assessment, CMOPL and PLOT—were classified as developing fundamental movement skills in early years aged children, i.e., 0–11 years old and 6 months–6 years old, respectively. This classification was established due to the fact that these tools are similarly characterised by the assessment of fundamental movement skills like stability, locomotion and manipulative skills. This finding is against the view of [31] who insist that the overall purpose of PL is to actively participate in physical activities for life.

Others on the other hand—CAPL, SHAPE America national standards for K-12 physical education and PE metrics, PPLI, CAEPL were identified as promoting long term health and well-being, i.e., free from hypokinetic diseases among the citizens particularly children and adolescents. The rationale behind this classification was the fact that these PL assessment tools put much emphasis on health related attributes: movement skills, fitness, living skills, physical activity and fitness, knowledge and skills, behaviour and value of physical activity. 

Such a philosophy of promoting health and well-being was partially matching with a predetermined overall purpose of physical literacy advanced by [32], which inculcates the participants to value and take the responsibility of participating in physical activity throughout the individual’s course of life.

In another respect, PFL and IPLA PL matrix were only identified as PL assessment tools developed under the holistic model, which takes into account development and promotion of PL under three domains—affective, physical and cognitive—and claimed to result in a desire to lifelong participation in physical activity. 

This finding was in agreement with the monism philosophy of physical literacy as explained by [7], that PL assessment tools will be appropriate when it has the ability to assess PL attributes under its three domains equally. Thus, human nature as a whole has to be taken into account in charting physical literacy journey. Additionally, this finding contradicts the assertion of [11], who states that a three-dimensional nature of PL—affective, physical and cognitive components—makes PL a challenge to wholly assess this concept using empirical tools.

An additional concern could be PLAY, which appears to holistically measure individuals’ PL progress. However, it has been criticised to treat such domains in a separate way and disproportionally focuses on physical competency [19,33]. Consistent with this contention, [32] comment that PLAY focuses much attention in fundamental movement skills, leading to fundamental sports skills of school aged children and adolescents in line of developing athletes and people able to participate in community activity. While [54] insist that PLAY attempts to assess confidence and comprehension (understanding), it is actually weak.

### 4.3. Who Have Been Targeted in Form of Age Groups

Thirdly, this review study sought to determine the classification of these PL assessment tools in relation to the target audience to be assessed in line of promoting their physical literacy. These review studies therefore found out that no PL assessment tool among these 10 was revealed to have a similar age group characteristics of the participants to be assessed. Thus, each tool targets a specific age group. 

To clarify, FMS: (60 MKC) Physical literacy assessment assesses PL of children aged 0–11 years; PLOT assesses children aged 6 months-6 years old; CAPL assesses children aged 8–12 years old, CAEPL assesses children aged 6 to 18 years old; PFL assesses children aged 6/7–18 years old; PLAY assesses 7–18 years old and above, SHAPE America assesses children aged 6/7–17/18 years old; PPLI assesses children aged 12/13–18/19 years old; CMOPL assesses children aged 6/7–17/18 years old and above. Nevertheless, a single IPLA physical literacy matrix was found as a tool designed to chart PL progress on an individual of any age group. 

This finding revealed a fact indicating that all existing PL assessment tools, except IPLA physical literacy matrix, were developed against the contention of [7] that physical literacy is philosophically founded respecting the nature of human being, whereby physical literacy considers every individual without concern of age group or of a living place. 

Such a finding was also against the view of [33], who revealed that nowadays PL initiatives focus their target mostly on children and adolescents against very little emphasis on pre-adolescents and adult people because of the fact established in this study that some tools assess PL among early years aged children (pre-adolescents) like (60 MKC) PL assessment and PLOT, while others such as PLAY and CMOPL assess PL of school aged children and adolescents and above, i.e., adults. 

### 4.4. How Is PL Assessment Being Undertaken 

The last objective of this study was to explore the assessment approaches adopted in these existing PL assessment tools when assessing/ charting PL progress of the concerned individuals. The study revealed that the majority—70% of the PL assessment tools presented in Table 1 including PLAY, CAPL, PFL, IPLA physical literacy matrix, CMOPL, CAEPL and PLOT, meet the characteristics of being classified in assessment for learning such as no comparison of PL level between participants, individualisation within the process of PL progress, collection of PL data overtime to monitor their PL journey, etc.

It is important to note at the same time that such a finding, asserted that assessment for learning or its combination with assessment *as* learning has been adopted to guide the process of assessing PL progress for both practitioners and participants using these PL assessment tools. 

This finding was consistent with the view of [4] that physical literacy is not concerned about learning experiences to be attained and maintained at a certain level, but a journey to be charted overtime. Along the same lines, it was in line with the phenomenology philosophy expressing the uniqueness of every individual and the view that the effective monitoring and promotion of PL should base on the previous individual knowledge, experience or understanding during physical literacy journey [7]. 

These data are also consistent with the view of [9], who elucidates that assessment *for* learning should be undertaken more than once by creating a description foregrounding every individual’s personal strengths and weaknesses to be used in serving the next stage of learning, and in turn provide the participants with necessary feedback that will further their learning. 

Notwithstanding the assessment for learning adopted, some remaining areas of improvement have been identified, for example: PFL compares students’ performance to standard of PL according to their age group not to each other [47] and comparison of students’ scores to their peers, a practice that is against the view of [14] that any comparison to any benchmark is inappropriate. 

It is important to note at the same time that SHAPE America was conceived quite irrelevant under this classification due to lack of phenomenological epistemology in its nature whereby it measures individuals’ PL progress against normative standards over a school year [25].

Another emerging issue to note is with regard to construct validity, whereby the findings show that the majority of the existing PL assessment tools still have a long way ahead for meeting a high level of construct validity. Notwithstanding that some PL assessment tools have never been tested for their level of validity, ref. [2,78] elucidate that a PL assessment tool which does not take into consideration multidimensional components of PL and measure it in a holistic approach remains barely classified valid. In light of this view, IPLA PL Matrix, PFL and CMOPL are interpreted valid. A view contradicting that of [31] who ascertained PFL a lowest level of validity.

Before drawing a conclusion, it is worth sounding a note of caution from the corresponding scholarly remarks such as fidelity to PL concept, usability, time consuming, exclusion of some age group as well as exclusion of disabled individuals, the consideration achieving PL literacy as an end point with a priority of normative standards and PL assessment approach to promote individuals’ PL journey. 

## 5. Future agenda

The following future agenda needs to be taken into consideration if physical literacy is understood as a journey to be charted for everybody—with any age group or living place, instead of a certain state of literacy to be attained and maintained thereafter.

It is worthwhile for the organisations, researchers and scholars designed or intending to design PL assessment tools to adjust or develop PL assessment tools appropriately fitting in the IPLA system of charting an individual’s progress on their physical literacy journey rather than assessing, measuring, evaluating, etc. 

For the sake of developing and promoting PL across the world, the future PL tools should be designed in a way providing opportunities for every one—of whatever age-group or living place—to charting his/her PL journey.

The organisations, policy makers, scholars or practitioners should link the process of charting PL progress with the definition and/or the existing attributes so as to signpost the correlation of such aspects of PL with participants’ practice. 

The concerned PL assessment developers or users should equally take into account both the philosophical principals—monism, existentialism, phenomenology and conceptual principals—motivation, confidence, physical competence, knowledge and understanding in charting an individual’s PL journey instead of allocating much emphasis on a single or a pair of PL domains. 

Since the majority of the existing PL assessment tools require the experts in physical activity to judge the participants’ practice, it is worth mentioning at this point that much consideration should be made in designing an unsophisticated tool enough to be completed by the participants by themselves, especially in the language used for descriptors (simple enough for participants to understand) and more importantly effective to be completed online. 

Lastly, the traditional assessment approach used to emphasise on assessment of learning with the rest small consideration of assessment for and as learning needs to be reconfigured into a preferred contemporary assessment approach that puts much emphasis on increasing consideration of assessment for and as learning while a relatively small assessment of learning comes out when there is a decision to be drawn just requiring summative judgement or when both teachers or learners wish to observe the cumulative effect of their work.

## 6. Conclusions

The study sought to scrutinise what is currently known regarding the PL assessment tools in relation to the philosophy behind their development, what they assess (assessment domains), the categories of population being assessed expresses in age group, as well as the approaches being used whether assessment for, as, or of learning within physical activity participation. 

The study established that the majority 70% of PL tools were developed to promote fundamental movement skills, long-term health and well-being, and athlete development instead of lifelong participation in physical activity. It was also ascertained that 30% of PL assessment tools address both three domains comprising PL. Of a particular concern, it was ascertained that only IPLA PL matrix takes into account everyone of any age group while the rest of others consider specific age group to be based on. Afterward, the majority of PL assessment tool were identified at 70% to adopt assessment for learning or at a certain time combination with assessment as learning while assessing individuals’ PL progress, a fact indicating that the majority of existing PL assessment tools still have areas of improvement so as to align with both Whitehead’s concept of PL as suggested by [79]. 

In view of these findings, the conclusion was therefore drawn that the overall purpose of PL—to value and take the responsibility of engaging in physical activity for life—is still largely absent among the majority of existing PL assessment tools, both the ideal of what to assess and who to be assessed are far less to be met, while the effective PL assessment approaches remain critical. Finally, in light of this conclusion, future agenda has been suggested in view of designing PL assessment tools effective enough to promote PL for all.

## Figures and Tables

**Figure 1 ijerph-18-07954-f001:**
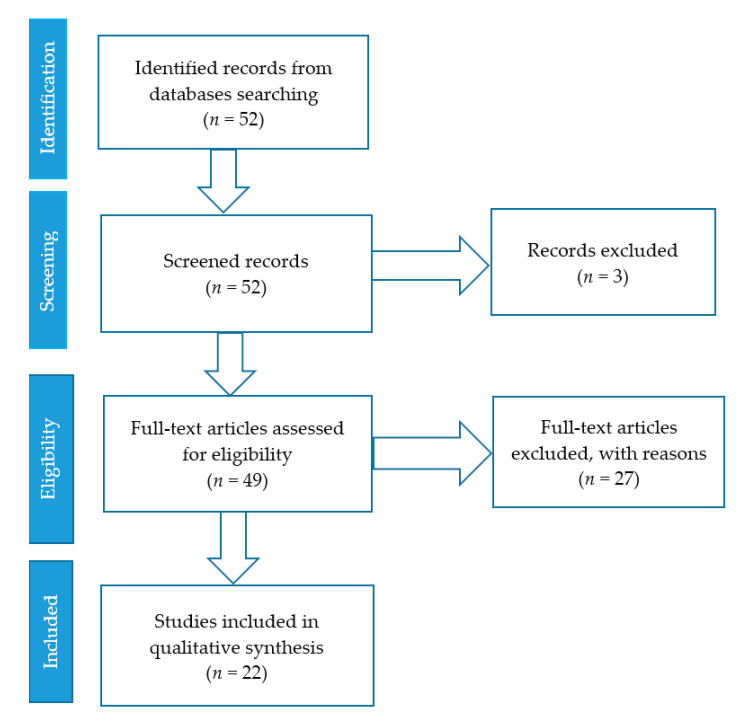
Flow chart of study selection process.

**Table 1 ijerph-18-07954-t001:** Physical literacy assessment tools.

Tool	Pioneer and Year	Assessment Domains	Target Audience Ages/Grades	Philosophy Adopted	Validity	Corresponding Scholarly Remarks
Passport For Life (PFL)	Professional organisation for physical and health educators of Canada PHE, Canada [47].	Movement skill test, fitness test, living skills, active participation.	School-aged children in grade K-12	Being and staying physically active for life (holistic development)	Lowest level of validity [31,48]	Prioritizing measures of physical domain-movement and fitness skills [34,49]; time consuming and exclusion of disabled individuals [50]; assessment of interaction with others [31]
Canadian Assessment for Physical Literacy (CAPL)	Healthy active living and obesity research group (HALO) [51]	Daily behaviour, motivation and confidence, physical competence, knowledge and understanding.	Children aged 8–12 years old	Long-term health and well-being	High level of validity [31,52].	Independent assessment of any PL domains, with standards and objectives allowing comparison among learners [53]; time consuming and exclusion of disabled individuals [50]
Physical Literacy Assessment for Youth (PLAY) tools	Kriellaars for Canadian Sport for Life (CS4L) [54]	Motor competence, knowledge, environmental participation.	Children and adolescence aged 7 years old and above	Athlete development and participation in community activity	Lowest level of validity [31,55]	Prioritization of physical competence through Long Term Athlete Development (LTAD) framework [31,34,56]; time consuming, assessors’ expertise and exclusion of disabled individuals [50]. Skill based [57].
FMS: 60 minutes Kids Club (60 MKC) Physical literacy assessment	Jupiter [58]	Fundamental movement skills (FMS): throwing, balance, catching, dodging, falling, galloping, hopping, jumping, kicking, object manipulation-hands and feet, rolling, running, starting and stopping, skipping, two hands striking	Children aged 0–11 years old	Development of fundamental movement skills in early years	_	The assessment of only single component/attribute of physical literacy [57].
IPLA Physical Literacy Matrix	Whitehead [7].	Motivation, confidence, physical confidence, knowledge and understanding.	Any individual of any age group	Being and staying physically active for life	_	Consideration of PL as a journey; interdependent domains of PL; individualisation in charting PL progress (phenomenology) [32].
Physical Literacy Observatory tool (PLOT)	Early years physical literacy research team [59]	Fundamental movement skills, stability skills, locomotor skills, and manipulative skills.	Children aged 6 months ±71 months	Development of fundamental movement skills in early years	_	The assessment of a single component/attribute of physical literacy [57].
National standards for K-12 physical education and PE metrics (SHAPE America)	Gu et al., USA [60]	Motor skills and movement patterns, movement and performance, knowledge, physical activity and fitness knowledge and skills, personal and social behaviour, and value physical activity.	Schools aged children and adolescence	Long term health and well being	_	Achievement of physical literacy as an end goal with a priority of normative standards and objectives. Thus, comparison between learners [31,53].
Perceived Physical Literacy Inventory (PPLI)	Sum et al. [61]	Knowledge and understanding, self-expression and communication, sense of self and self-confidence.	Adolescents	Long term health and well being	High level of validity [62]	Not designed for a specific population or profession [62]
Chinese Assessment and evaluation of physical literacy (CAEPL)	Shanghai University Sport [57]	Intention of physical activity, knowledge of physical activity, motor/sport skills, behaviour of physical activity, physical fitness.	Children aged 6–18 years old	Long term health and well being	_	Inequality treatment of all domains (largest weight to motor skills, directing much attention to behaviour and skills); still theoretical model [63].
Conceptual model of observed physical literacy(CMOPL)	Dudley, Australia [32]	Movement competencies; rules, tactics and strategies of movement, motivation and behavioural skills of movement, personal and social attributes of movement.	School-aged children and beyond school	Being and staying physically active for life	_	Directing attention to PL as journey to be charted overtime; interrelated domains of PL; individual based PL progress charting [32].

Source: Researcher 2021.

**Table 2 ijerph-18-07954-t002:** Physical literacy assessment tools and their corresponding guiding PL definitions.

PL Assessment Tool	PL Definition
Passport for Life (PFL)	An individual who is moving with competence and confidence in a wide variety of physical activities in multiple environments that benefit the health development of the whole person {[48] (p. 442), [64]}.
Canadian assessment for physical literacy (CAPL)	The motivation, confidence, physical competence knowledge and understanding to value and take responsibility for engagement in physical activities for life [65] (p. 1).
Physical literacy assessment for youth tools (PLAY)	Individuals are physically literate when they have acquired the movement skills and confidence to enjoy a variety of sports and physical activities {[66] (p. 4), [67] (p. 1)}.
FMS: 60 minutes Kids Club (60 MKC) PL assessment	The motivation, confidence, physical competence, knowledge and understanding to value and take responsibility of engagement in physical activities for life [58].
IPLA physical literacy matrix	The motivation, confidence, physical competence, knowledge and understanding to value and take responsibility of engaging in physical activity for life [7] (p. 8).
Physical literacy observatory tool (PLOT)	The motivation, confidence and competence to move for a lifetime [59].
National standards for K-12 physical education and PE metrics (SHAPE America)	Physically literate individuals-have the knowledge, skills and confidence to enjoy a lifetime of healthful physical activity [68].
Perceived physical literacy inventory (PPLI)	A specific intelligence that includes the motivation, confidence, physical competence, knowledge and understanding to value and take responsibility for maintaining purposeful physical pursuits and activities throughout the course of one’s life [62] (p. 27).
Chinese assessment and evaluation of physical literacy (CAEPL)	A comprehensive capability integrating different components that benefit individual active lifestyles and health throughout life span [69].
Conceptual model of observed physical literacy (CMOPL)	The lifelong holistic learning acquired and applied in movement and physical activity which brings together the skills, capabilities and knowledge which we know contribute to well-rounded people who value and participate in an active life [70].

Source: Researcher (2021).

## Data Availability

Not applicable.

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
