# Peer review of "Physical Literacy Assessment Tools: A Systematic Literature Review for Why, What, Who, and How"

_ijerph, 2021, doi:10.3390/ijerph18157954_

Round 1

Reviewer 1 Report

The systematic literature review about Physical Literacy assessment tools is current and relevant.

Need to modify: the lack of consensus on the definition of Physical Literacy. The different philosophical approaches used and the tested theories give the manuscript the necessary criteria to be considered useful and up-to-date. In light of these topics, there is great controversy over whether physical literacy can be measured.

Since the aim of the study was to examine what is currently known about Physical Literacy assessment tools and in light of the entire analysis of the results, the authors should state the key points of the study.

Congratulations for the manuscript.

Author Response

thank you so much for your comment, actually the key points of the study have been stated in Abstract and introduction but the authors addressed this comment in the manuscripts by clarifying  all the points. these are: the existing PL assessment tools, their pioneers & year of publication, the philosophy behind their initiation, what they have been assessing (assessment domains), the category of population being assessed in relation to their age group, validity of the tools, other scholars notes, as well as approaches being used whether assessment for, as, or of learning during physical activity participation.

Reviewer 2 Report

The article is well written and presented

Why was it chosen to exclude articles prior to 2000?

I agree with the idea of the authors, there should present a version of PL evaluation updated, possibly uniform and continuos; in this sense, they should present at least hypothetical guidelines to draft it

The reason for the exclusion of the 10 studies is not clear

Could you analyze the difference in both evaluation and final results in different countries or continents?

Could you give your definition of PL?

Line 198 "fourth"

Line 385 no capital letter needed

Author Response

Thank you so much for your comments,

The reason behind the exclusion of articles published prior to year 2000 was the fact that consideration of last 20 years for a new concept which lasts less than three decades is enough to ascertain its current development particularly assessment mechanisms.

the reason of excluding 10 articles is inserted in the manuscript that it was due to the fact that they didn't describe the assessment mechanisms of the tool(s) but addressing the tool(s) adopted in study to assess PL in their interventions.

concerning the analysis about difference in both evaluation and final results in different countries or continent, the authors considered it not applicable. The reason could be the fact that all continents or countries are not represented considering the results of the study. Thus, such an analysis could introduce a slight bias. 

the definition of PL according to the authors could be: PL refers to a monism, existentialism & phenomenology philosophical based disposition conceived to curb global physical inactivity while increasing lifelong participation in physical activity by promoting the motivation & confidence, knowledge & understanding and physical competence of all people irrespective of their age group or living place. 

the required revisions reflected in line 198 and 385 are all addressed.